# Favipiravir, lopinavir-ritonavir, or combination therapy (FLARE): A randomised, double-blind, 2 × 2 factorial placebo-controlled trial of early antiviral therapy in COVID-19

**David M. Lowe**[1,2☯]*, **Li-An K. Brown**[1☯], **Kashfia Chowdhury**[3], **Stephanie Davey**[4], **Philip Yee**[4], **Felicia Ikeji**[3], **Amalia Ndoutoumou**[3], **Divya Shah**[5], **Alexander Lennon**[5], **Abhulya Rai**[5], **Akosua A. Agyeman**[6], **Anna Checkley**[7], **Nicola Longley**[7], **Hakim-Moulay Dehbi**[3], **Nick Freemantle**[3], **Judith Breuer**[5,6‡], **Joseph F. Standing**[6,8‡], **FLARE Investigators**[¶]

1 Institute of Immunity and Transplantation, University College London, London, United Kingdom,
2 Department of Clinical Immunology, Royal Free London NHS Foundation Trust, London, United Kingdom,
3 Comprehensive Clinical Trials Unit, University College London, London, United Kingdom, 4 Department of Rheumatology, Royal Free London NHS Foundation Trust, London, United Kingdom, 5 Department of Virology, Great Ormond Street Hospital NHS Foundation Trust, London, United Kingdom, 6 Infection, Immunity and Inflammation Research and Teaching Department, Institute of Child Health, University College London, London, United Kingdom, 7 University College London Hospitals NHS Foundation Trust, London, United Kingdom, 8 Department of Pharmacy, Great Ormond Street Hospital NHS Foundation Trust, London, United Kingdom

☯ These authors contributed equally to this work.
‡ JB and JFS also contributed equally to this work.
¶ Membership of FLARE Investigators is provided in S1 Appendix.
* d.lowe@ucl.ac.uk

**Data Availability Statement:** All relevant data are within the manuscript and its Supporting Information files. Patient level data are available from the Clinical Director of the UCL Comprehensive Clinical Trials Unit (a.o'brien@ucl.

## Abstract

### Background

Early antiviral treatment is effective for Coronavirus Disease 2019 (COVID-19) but currently available agents are expensive. Favipiravir is routinely used in many countries, but efficacy is unproven. Antiviral combinations have not been systematically studied. We aimed to evaluate the effect of favipiravir, lopinavir-ritonavir or the combination of both agents on Severe Acute Respiratory Syndrome Coronavirus 2 (SARS-CoV-2) viral load trajectory when administered early.

### Methods and findings

We conducted a Phase 2, proof of principle, randomised, placebo-controlled, 2 × 2 factorial, double-blind trial of ambulatory outpatients with early COVID-19 (within 7 days of symptom onset) at 2 sites in the United Kingdom. Participants were randomised using a centralised online process to receive: favipiravir (1,800 mg twice daily on Day 1 followed by 400 mg 4 times daily on Days 2 to 7) plus lopinavir-ritonavir (400 mg/100 mg twice daily on Day 1, followed by 200 mg/50 mg 4 times daily on Days 2 to 7), favipiravir plus lopinavir-ritonavir placebo, lopinavir-ritonavir plus favipiravir placebo, or both placebos. The primary outcome

ac.uk) upon reasonable request. This is in line with the informed consent form for this study where participants consented for their data to be shared with other researchers only for ethically approved research following legal requirements to conceal their identity.

**Funding:** This work was supported by LifeArc (www.lifearc.org; grant COVID0005 to DML) and the UK Medical Research Council (MRC, www.ukri.org/councils/mrc; fellowship MR/M008665/ to JFS and project grant MR/W015560/1 to AAA). The funders had no role in study design, data collection and analysis, decision to publish, or preparation of the manuscript but LifeArc were provided a copy of the manuscript before submission.

**Competing interests:** I have read the journal's policy and the authors of this manuscript have the following competing interests: DML has received personal fees from Gilead for an educational video on COVID-19 in immunodeficiency and from Merck for a roundtable discussion on risk of COVID-19 in immunosuppressed patients. DML also holds research grants from Blood Cancer UK, Bristol Myers Squibb and the British Society for Antimicrobial Chemotherapy, all outside the current work. NF has received funding from Gedeon Richter, Abbott Singapore, Galderma, ALK, AstraZeneca, Ipsen, Vertex, Novo Nordisk, Aimmune, Allergan and Novartis, all outside the current work. JB holds research funding from GSK, Wellcome Trust, UKRI, Rosetrees Foundation and the John Black Foundation, all outside the current work. All other authors declare no conflict of interest.

**Abbreviations:** ANCOVA, analysis of covariance; BMI, body mass index; CI, confidence interval; COVID-19, Coronavirus Disease 2019; Ct, cycle threshold; GOSH, Great Ormond Street Hospital; IDMC, Independent Data Monitoring Committee; ITT, intention-to-treat; MERS-CoV, Middle East Respiratory Syndrome coronavirus; mITT, modified intention-to-treat; NHS, National Health Service; OR, odds ratio; PCR, polymerase chain reaction; RdRp, ribosomal-dependent RNA polymerase; SARS-CoV-2, Severe Acute Respiratory Syndrome Coronavirus 2; UCLH, University College London Hospital; UPH, Urgent Public Health.

was SARS-CoV-2 viral load at Day 5, accounting for baseline viral load. Between 6 October 2020 and 4 November 2021, we recruited 240 participants. For the favipiravir+lopinavir-ritonavir, favipiravir+placebo, lopinavir-ritonavir+placebo, and placebo-only arms, we recruited 61, 59, 60, and 60 participants and analysed 55, 56, 55, and 58 participants, respectively, who provided viral load measures at Day 1 and Day 5. In the primary analysis, the mean viral load in the favipiravir+placebo arm had changed by $-0.57$ $\log_{10}$ (95% CI $-1.21$ to 0.07, $p = 0.08$) and in the lopinavir-ritonavir+placebo arm by $-0.18$ $\log_{10}$ (95% CI $-0.82$ to 0.46, $p = 0.58$) compared to the placebo arm at Day 5. There was no significant interaction between favipiravir and lopinavir-ritonavir (interaction coefficient term: 0.59 $\log_{10}$, 95% CI $-0.32$ to 1.50, $p = 0.20$). More participants had undetectable virus at Day 5 in the favipiravir+placebo arm compared to placebo only (46.3% versus 26.9%, odds ratio (OR): 2.47, 95% CI 1.08 to 5.65; $p = 0.03$). Adverse events were observed more frequently with lopinavir-ritonavir, mainly gastrointestinal disturbance. Favipiravir drug levels were lower in the combination arm than the favipiravir monotherapy arm, possibly due to poor absorption. The major limitation was that the study population was relatively young and healthy compared to those most affected by the COVID-19 pandemic.

## Conclusions

At the current doses, no treatment significantly reduced viral load in the primary analysis. Favipiravir requires further evaluation with consideration of dose escalation. Lopinavir-ritonavir administration was associated with lower plasma favipiravir concentrations.

## Trial registration

Clinicaltrials.gov NCT04499677
EudraCT: 2020-002106-68

## Author summary

### Why was this study done?

* The FLARE trial aimed to discover whether existing oral antiviral drugs could reduce the viral load of the Severe Acute Respiratory Syndrome Coronavirus 2 (SARS-CoV-2) virus if given soon after symptoms started.

* If effective this strategy could reduce the risk of hospitalisation and death from Coronavirus Disease 2019 (COVID-19).

### What did the researchers do and find?

* The researchers performed a clinical trial of 2 medications—favipiravir and lopinavir/ritonavir, testing them on their own and in combination.

* Combination therapies were less effective than favipiravir monotherapy, but many people taking lopinavir/ritonavir had gastrointestinal side effects and favipiravir drug levels were lower in the combination arm, possibly due to poor absorption.

* SARS-CoV-2 viral loads were not significantly lower with any of the drug treatments after 5 days compared to placebo, although more people taking favipiravir had undetectable levels of the virus.

### What do these findings mean?

* None of these therapies should be used routinely at the current doses investigated.

* Further studies investigating the effect of favipiravir when administered at higher doses should be undertaken.

## Introduction

Severe Acute Respiratory Syndrome Coronavirus 2 (SARS-CoV-2) continues to represent a major threat to global health. Interrupting viral replication in early infection reduces the risk of Coronavirus Disease 2019 (COVID-19) disease progression and hospitalisation [1–5]. Efficacy has been demonstrated for neutralising monoclonal antibody treatments, but these are vulnerable to loss of potency with new viral variants as observed with the B.1.1.529 (omicron) variant [6]. Furthermore, the cost of available oral antiviral and monoclonal treatments is prohibitive for many countries.

A general principle of antiviral chemotherapy is that multiple agents with different modes of action are often required, which can be particularly pertinent in the case of repurposed drugs where antiviral potency using monotherapy may be limited. Combination therapy using a polymerase inhibitor combined with a protease inhibitor, thereby targeting sequential steps in the viral replication pathway, is a potential strategy [7]. When SARS-CoV-1 was treated with the polymerase inhibitor ribavirin in combination with the protease inhibitor lopinavir-ritonavir, and when this combination was initiated immediately upon diagnosis, a significantly lower mortality was seen compared with historical controls [8]. Another study of this combination showed reduced mortality and need for intubation when therapy was given early, while late rescue treatment had no effect [9]. Early post-exposure prophylaxis against Middle East Respiratory Syndrome (MERS-CoV) in healthcare workers also showed that ribavirin plus lopinavir-ritonavir reduced the incidence of infection from 28% to 0% [10].

In early 2020, it was shown that while ribavirin had little effect on SARS-CoV-2 viral replication in vitro, the orally available polymerase inhibitor favipiravir did have an in vitro potency within clinically achievable range [11] (S1 Fig). While subsequent in vitro results have been less promising, high-dose favipiravir achieving concentrations commensurate with human exposures reduced viral load and lung pathology in hamsters [12]. Early observational clinical studies reported benefits of favipiravir in COVID-19 patients [13,14]. Favipiravir generic formulations are now in widespread use for COVID-19 in some countries, but high-quality evidence on its effect in early treatment is lacking. A recent pre-print suggested that favipiravir (as monotherapy and taken with a twice daily dosing regimen) did not impact time to viral clearance [15].

While the HIV protease inhibitors tipranavir and nelfinavir showed higher in vitro potency against SARS-CoV-2 than lopinavir-ritonavir [11], safety concerns and limited clinical experience with these agents meant that we chose to study lopinavir-ritonavir. Both lopinavir and ritonavir, which is used as a pharmacokinetic booster to lopinavir, have modest anti-

SARS-CoV-2 activity in vitro [11] that was predicted to yield around up to 30% inhibition of viral replication at the licensed dose (S1 Fig). In line with this, lopinavir-ritonavir monotherapy did not improve clinical outcomes in trials on hospitalised patients [16,17]. However, viral dynamic modelling suggests that drugs with lower potency may nevertheless inhibit viral replication if started earlier [18,19], and high-quality early treatment trials with lopinavir-ritonavir are lacking.

The FLARE trial therefore aimed to deliver robust Phase 2, proof of principle, data on viral load changes using early antiviral treatment. The combination of favipiravir plus lopinavir-ritonavir was studied in a 2 × 2 factorial design to evaluate the combination while simultaneously testing each agent in monotherapy versus placebo to understand their respective contributions. Doses used in current clinical practice and previous trials for other indications were used due to available safety data, and modelling which suggested that we would achieve EC90 for favipiravir based on the available pharmacokinetic data at the time (S1 Fig). For favipiravir, this is similar to the dose now being employed worldwide for COVID-19.

## Materials and methods

### Study design and participants

FLARE was an early intervention trial testing the effect of oral antiviral therapy on viral load [20] in ambulatory outpatients. Participants received favipiravir plus lopinavir-ritonavir, favipiravir plus lopinavir-ritonavir placebo, favipiravir placebo plus lopinavir-ritonavir, or placebos of both drugs. Favipiravir or matched placebo was administered at a dose of 1,800 mg twice daily on Day 1, followed by 400 mg 4 times daily from Day 2 to Day 7. Lopinavir-ritonavir or matched placebo were given at a dose of 400 mg/100 mg twice daily on Day 1, followed by 200 mg/50 mg 4 times daily from Day 2 to Day 7. Participants were advised to take both Day 1 doses on the first day regardless of time of enrolment, due to the perceived importance of achieving high antiviral levels as early as possible. Those recruited in the afternoon took the first dose immediately and the second dose at least 6 hours later.

Participants aged between 18 and 70 years who had recently (within the last 5 days) developed symptoms of COVID-19, or who had tested positive for SARS-CoV-2 by polymerase chain reaction (PCR) and were within 7 days of symptom onset, or who were asymptomatic but had tested positive by PCR within the previous 48 hours, were eligible for the trial. Participants were ineligible if they had known hypersensitivity to either drug or their ingredients/ excipients, had chronic liver or kidney disease, were taking concomitant medicines known to interact with the trial treatments, were being treated as a hospital inpatient for any condition, were pregnant or breastfeeding, or were participating in another interventional clinical trial (treatment or vaccination). Before 8 June 2021, participants vaccinated against SARS-CoV-2 were excluded but this was reversed by the Trial Steering Committee due to the large number of vaccinated individuals presenting with infection at that time, and the importance of establishing whether early antiviral treatment is effective in a vaccinated population. Female participants of childbearing potential were required to provide a negative pregnancy test before commencement of trial medication and on Day 14, and to use highly effective contraceptive measures during the trial; male participants with a female partner of childbearing potential were also required to use highly effective contraception.

Participants were informed about the trial via occupational health departments at participating hospital sites and participant identification centres, via poster advertisements, social media or, from 23 June 2021, directly by National Health Service (NHS) Test and Trace following the identification of a positive test. The trial team also directly contacted ambulatory

patients who had tested positive at hospital sites and those in the local area from a list provided by NHS Digital.

Participants were recruited at 2 sites: Royal Free Hospital and University College London Hospital, both in London, UK.

The study was approved by the Wales Research Ethics Committee 3 (Ref: 20/WA/0210) and all participants provided written, informed consent. The trial registration number (clinicaltrials.gov) was NCT04499677.

## Randomisation and masking

A prescreening visit (usually by telephone) briefly assessed eligibility and collected the following information: study site, age ($\leq$55 versus >55 years), sex, height and weight (to calculate body mass index (BMI)), symptomatic or asymptomatic, current smoking status (current smoker, non-smoker, ex-smoker), ethnicity, previous COVID-19 specific vaccination (yes/no), and presence/absence of the following comorbidities: diabetes, hypertension, ischaemic or other heart disease, and chronic respiratory disease. These variables were used as part of the minimisation strategy to randomise participants into the 4 arms 1:1:1:1 using a centralised concealed online process to assign participants to a medication kit number.

Trial medication kits, prepared by RenaClinical, were coded to maintain double blinding (investigators and participants). Kits contained favipiravir or colour and size matched placebo 200-mg tablets supplied by Fujifilm Toyama Chemical Co. and lopinavir-ritonavir 200-mg/50-mg tablets (AbbVie) or colour and size matched placebos (RenaClinical).

## Study procedures

People willing to participate at prescreening were visited in their home or seen in a designated COVID-19 treatment area at recruitment sites. Following confirmation of eligibility and written informed consent, a nasopharyngeal swab (for participants who were symptomatic but had not tested positive) and baseline blood test was performed along with collection of clinical and demographic information. A pack containing trial medication, kits and instructions for collecting daily saliva samples (Saliva RNA Collection and Preservation devices, Norgen Biotek, Canada), a thermometer, and participant diary was provided. The first saliva sample was taken followed by witnessed intake of the first dose of trial medication; participants were advised to take daily saliva samples each morning from Days 2 to 7 before eating, drinking, or brushing teeth.

A telephone follow-up was performed on Day 5 and a second visit performed on Day 7 where saliva samples were collected and blood was drawn for safety and favipiravir pharmacokinetics. Stool samples were collected if provided. Follow-up telephone calls or visits were made on Day 14; a pregnancy test was performed for women of childbearing potential and blood tests taken if abnormalities had been detected at Day 7. A final telephone call was made on Day 28.

## Outcomes

The primary outcome was viral load measured by quantitative PCR performed on saliva samples at Day 5, accounting for the pretreatment Day 1 viral load. Secondary outcomes were proportion of participants with undetectable viral loads at Day 5, rate of decrease in viral load during the 7-day treatment course, duration of fever, proportion of participants with medication-related toxicity at Days 7 and 14, and proportion of participants admitted to hospital, intensive care, or dead due to a COVID-19-related illness.

We planned to assess viral clearance in stool but received insufficient samples for analysis. Further outcomes of whole-genome sequencing of SARS-CoV-2 and more extensive pharmacokinetic-pharmacodynamic modelling will be reported separately.

## Laboratory analyses

Full blood count, urea and electrolytes, liver function tests, and serum urate were measured in the diagnostic laboratory at Great Ormond Street Hospital (GOSH), London. Saliva viral load was also measured by the GOSH diagnostic laboratory. Samples with a cycle threshold (Ct) value between 40 to 45 were repeated, and for the purposes of the primary analysis, a viral load was calculated from the calibration curve if the repeat value was also <45. However, due to uncertainties in the interpretation of these Ct values and in line with clinical practice, for the secondary analysis of undetectable viral load, Ct values >40 were considered undetectable.

Serum antibody status at Day 1 and Day 7 was measured at the University of Birmingham via enzyme-linked immunosorbent assay, as described previously [21].

Favipiravir drug levels pre and post the second or third dose on Day 7 were measured in plasma by the LSI Medience Corporation in Japan on behalf of Fujifilm Toyama Chemical Co. Favipiravir was confirmed to be stable for 24 hours at room temperature and for 6 months once frozen at −20°C. The assay lower limit of quantification was 0.1 mg/L.

## Statistical analysis

It was assumed that a clinically significant difference in viral load between antiviral and placebo-treated participants would be 0.5 to 1 $\log_{10}$ copies/mL by Day 5. Simulations showed a total of 216 participants would provide 90% power with 2-sided alpha of 2.5% to detect a 0.9 $\log_{10}$ decrease in viral load of each active treatment on its own compared to placebo. The factorial design allowed an interaction term to be estimated with 80% power, at a nominal 2-sided alpha of 5%, to detect a synergistic or antagonistic effect of 1.0 $\log_{10}$ copies/mL. The standard deviation of change in viral load in each group was assumed to be 1.3 $\log_{10}$ copies/mL. To allow for 10% attrition rate, a total sample size of 240 (60 participants per arm) was determined.

All statistical analyses were done according to a predefined statistical analysis plan (see S1 Statistical Analysis). Analysis of the primary, secondary, and safety outcomes was conducted on the intention-to-treat (ITT) population. The ITT population is composed of all randomised participants. For the primary outcome, the ITT analysis was composed of all ITT participants for whom a measure of viral load was available at Day 1 and Day 5. Additionally, the primary outcome was analysed in a modified ITT (mITT) population, which excluded participants who had undetectable viral load both at Day 1 and Day 5.

An analysis of covariance (ANCOVA) model was used to estimate the difference in viral load at 5 days post treatment between the treatment arms. The model included a term for each treatment (favipiravir active/placebo and lopinavir-ritonavir active/placebo), an interaction term between the 2 treatments, and baseline viral load. Supportive analyses on the primary outcome included a model adjusting for (i) minimisation factors; (ii) minimisation factors, symptom duration and antibody status (post hoc adjustment strategy); (iii) potential effect of the delta variant of the SARS-CoV-2 virus, by adding a categorical variable reflecting the period of recruitment: no delta variant (before 24 April 2021), some delta variant (between 24 April 2021 and 12 June 2021) and predominantly delta variant period (post 12 June 2021). A linear mixed model was used to model the viral load trajectories from Day 1 to Day 7 between the 4 treatment arms. Two adjustment strategies were followed: (i) Day 2 to Day 7 viral loads were modelled as response variable, adjusted for Day 1 viral load; (ii) also adding minimisation

factors, symptom duration, and antibody status (post hoc analysis). Logistic regression models were fitted to test for differences in proportions of events between the groups. We used STATA/MP 17.0 for all analyses.

No interim analyses were planned and safety monitoring was undertaken by an Independent Data Monitoring Committee (IDMC).

## Results

### Study participants

Between 6 October 2020 and 4 November 2021, we screened 1,215 and recruited 240 participants (Fig 1). Participant details are provided in Table 1 and minimisation factors in Table 2. Most participants (90%) were below the age of 55 years; 82% were Caucasian and 85% did not have any comorbidities. Approximately 51% of those randomised were vaccinated against SARS-CoV-2, and the proportion of vaccinated participants was balanced across the 4 arms;

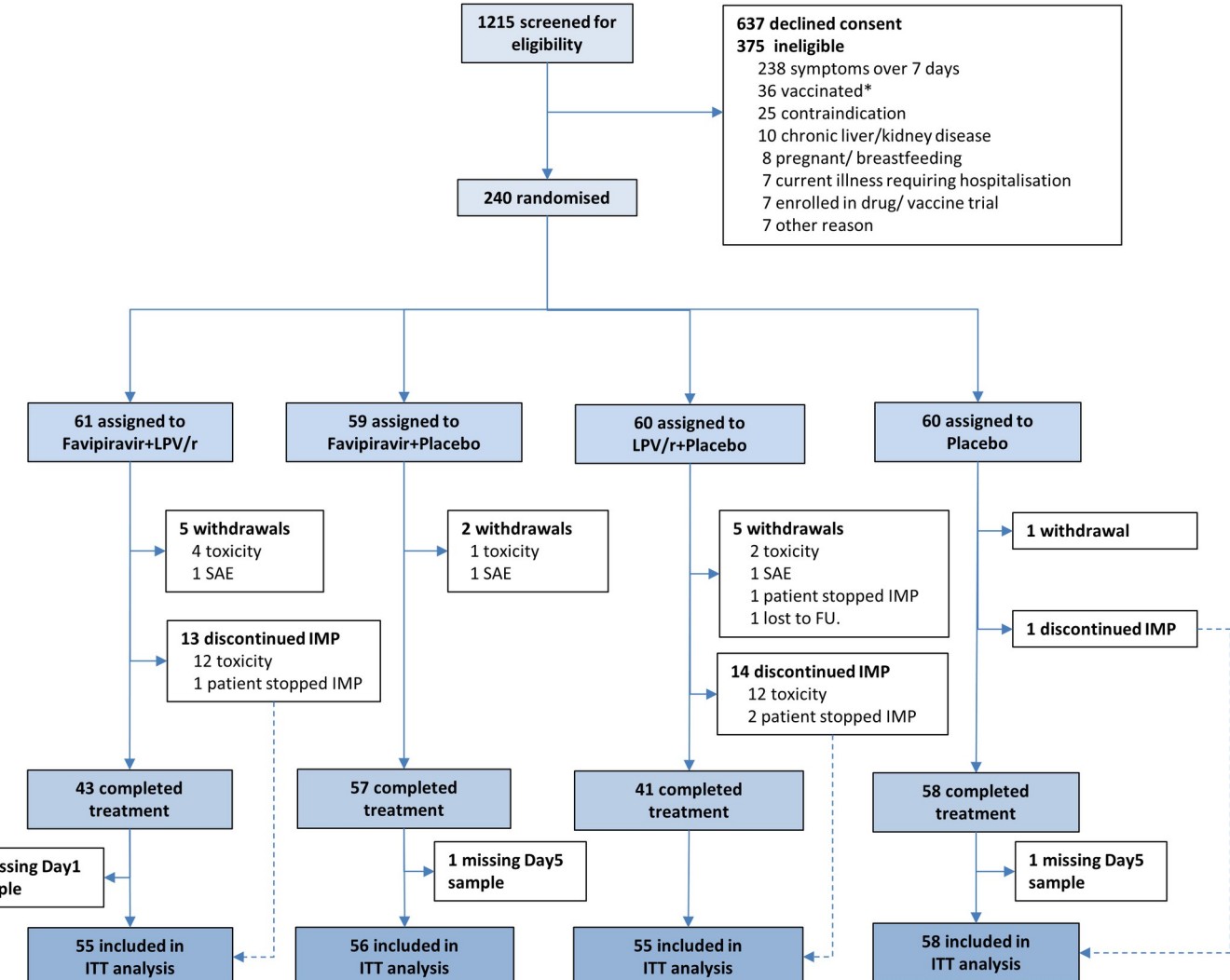

**Fig 1. CONSORT diagram for the FLARE trial.** * SARS-CoV-2 vaccination was an exclusion in the earlier part of the trial. FU, follow-up; IMP, investigational medicinal product; ITT, intention-to-treat LPV/r, lopinavir-ritonavir; SAE, serious adverse event; SARS-CoV-2, Severe Acute Respiratory Syndrome Coronavirus 2.

**Table 1. Participant baseline characteristics.**

| Characteristics at screening | | Favipiravir+LPV/r (N = 61) | Favipiravir+Placebo (N = 59) | LPV/r+Placebo (N = 60) | Placebo (N = 60) | Total (N = 240) |
|---|---|---|---|---|---|---|
| Age (years) | mean (sd) | 40.3 (13.1) | 40.3 (12.1) | 38.6 (11.5) | 40.6 (12.2) | 40.0 (12.2) |
| Height (cm) | mean (sd) | 172.8 (9.1) | 172.5 (9.6) | 172.1 (9.7) | 171.2 (9.7) | 172.2 (9.5) |
| Weight (kg) | mean (sd) | 76.0 (17.0) | 76.5 (14.1) | 74.8 (16.6) | 75.4 (15.9) | 75.7 (15.9) |
| Pulse rate (bpm) | mean (sd) | 72.6 (11.4) | 72.6 (11.1) | 76.9 (10.5) | 75.2 (10.9) | 74.3 (11.1) |
| Respiratory rate (bpm) | mean (sd) | 16.9 (3.5) | 16.5 (2.6) | 16.6 (2.7) | 16.8 (2.8) | 16.7 (2.9) |
| Body temperature (°C) | mean (sd) | 36.8 (0.7) | 36.7 (0.6) | 36.8 (0.7) | 36.6 (0.6) | 36.7 (0.6) |
| HIV status | n (%) | | | | | |
| Positive | | 1 (1.6) | 0 (0.0) | 0 (0.0) | 0 (0.0) | 1 (0.4) |
| Negative | | 17 (27.9) | 22 (37.3) | 20 (33.3) | 19 (31.7) | 78 (32.5) |
| Unknown | | 43 (70.5) | 37 (62.7) | 40 (66.7) | 41 (68.3) | 161 (67.1) |
| Vaccinated | n (%) | | | | | |
| Yes | | 32 (52.5) | 30 (50.8) | 31 (51.7) | 30 (50.0) | 123 (51.2) |
| No | | 29 (47.5) | 29 (49.2) | 29 (48.3) | 30 (50.0) | 117 (48.8) |
| Type of vaccine | n (%) | | | | | |
| Pfizer/BioNTech | | 14 (23.0) | 13 (22.0) | 19 (31.7) | 8 (13.3) | 54 (22.5) |
| Oxford/AstraZeneca | | 16 (26.2) | 17 (28.8) | 12 (20.0) | 21 (35.0) | 66 (27.5) |
| Moderna | | 2 (3.3) | 0 (0.0) | 0 (0.0) | 1 (1.7) | 3 (1.3) |
| Number of doses | n (%) | | | | | |
| One | | 7 (11.5) | 5 (8.5) | 7 (11.7) | 3 (5.0) | 22 (9.2) |
| Two | | 24 (39.3) | 25 (42.4) | 24 (40.0) | 27 (45.0) | 100 (41.7) |
| Three | | 1 (1.6) | 0 (0.0) | 0 (0.0) | 0 (0.0) | 1 (0.4) |
| Symptom onset | n (%) | | | | | |
| ≤5 days | | 43 (70.5) | 39 (66.1) | 38 (63.3) | 37 (62.7) | 157 (65.7) |
| >5 days | | 18 (29.5) | 20 (33.9) | 22 (36.7) | 22 (37.3) | 82 (34.3) |
| Symptoms | n (%) | | | | | |
| Fever | | 32 (52.5) | 23 (38.3) | 33 (54.1) | 35 (58.3) | 123 (50.8) |
| Cough | | 48 (78.7) | 36 (60.0) | 42 (68.9) | 43 (71.7) | 169 (69.8) |
| Anosmia | | 0 (0.0) | 0 (0.0) | 1 (1.67) | 0 (0.0) | 1 (1.67) |
| Shortness of breath | | 10 (16.4) | 13 (21.7) | 8 (13.1) | 12 (20.0) | 43 (17.8) |
| Malaise | | 37 (60.7) | 36 (60.0) | 38 (62.8) | 28 (46.7) | 139 (57.4) |
| Myalgia | | 34 (55.7) | 33 (55.0) | 32 (52.5) | 28 (46.7) | 127 (52.5) |
| Headache | | 34 (55.7) | 40 (66.7) | 36 (59.0) | 33 (55.0) | 143 (59.1) |
| Coryza | | 22 (36.1) | 23 (38.3) | 27 (44.3) | 20 (33.3) | 92 (38.0) |
| Other | | 42 (68.9) | 45 (75.0) | 49 (80.3) | 39 (65.0) | 175 (72.3) |
| Antibody status | n (%) | | | | | |
| Negative | | 21 (34.4) | 21 (36.2) | 23 (38.3) | 23 (38.3) | 88 (36.8) |
| Positive | | 40 (65.6) | 37 (63.8) | 37 (61.7) | 37 (61.7) | 151 (63.2) |

bpm, beats per minute; LPV/r, lopinavir-ritonavir.

63% had detectable SARS-CoV-2 anti-spike antibody at baseline. Approximately 66% of the participants started treatment within 5 days of symptom onset. The time between symptom onset and start of treatment was similar between the arms. Three patients did not have a positive SARS-CoV-2 PCR prior to recruitment; of these, 2 had a positive baseline sample but 1 had a negative baseline sample. One patient was asymptomatic at recruitment but their baseline viral load was above the mean. Symptoms across the entire cohort are summarised in S1 Table.

**Table 2. Participant minimisation factors.**

| Minimisation factors N (%) | Favipiravir+LPV/r (N = 61) | Favipiravir+Placebo (N = 59) | LPV/r+Placebo (N = 60) | Placebo (N = 60) | Total (N = 240) |
|---|---|---|---|---|---|
| Site | | | | | |
| Royal Free | 56 (91.8) | 55 (93.2) | 55 (91.7) | 55 (91.7) | 221 (92.1) |
| UCLH | 5 (8.2) | 4 (6.8) | 5 (8.3) | 5 (8.3) | 19 (7.9) |
| Age (years) | | | | | |
| ≤55 | 53 (86.9) | 52 (88.1) | 55 (91.7) | 55 (91.7) | 215 (89.6) |
| >55 | 8 (13.1) | 7 (11.9) | 5 (8.3) | 5 (8.3) | 25 (10.4) |
| Gender | | | | | |
| Male | 31 (50.8) | 32 (54.2) | 29 (48.3) | 31 (51.7) | 123 (51.2) |
| Female | 30 (49.2) | 27 (45.8) | 31 (51.7) | 29 (48.3) | 117 (48.8) |
| Ethnicity | | | | | |
| Caucasian | 50 (82.0) | 49 (83.1) | 49 (81.7) | 49 (81.7) | 197 (82.1) |
| Other | 11 (18.0) | 10 (16.9) | 11 (18.3) | 11 (18.3) | 43 (17.9) |
| BMI (kg/m$^2$) | | | | | |
| <30 | 51 (83.6) | 49 (83.1) | 50 (83.3) | 50 (83.3) | 200 (83.3) |
| ≥30 | 10 (16.4) | 10 (16.9) | 10 (16.7) | 10 (16.7) | 40 (16.7) |
| Symptomatic disease | | | | | |
| Yes | 61 (100.0) | 59 (100.0) | 60 (100.0) | 59 (98.3) | 239 (99.6) |
| No | 0 (0.0) | 0 (0.0) | 0 (0.0) | 1 (1.7) | 1 (0.4) |
| Current smoker | | | | | |
| Yes | 6 (9.8) | 7 (11.9) | 7 (11.7) | 7 (11.7) | 27 (11.3) |
| No | 55 (90.2) | 52 (88.1) | 53 (88.3) | 53 (88.3) | 213 (88.8) |
| Vaccinated | | | | | |
| Yes | 32 (52.5) | 30 (50.8) | 31 (51.7) | 30 (50.0) | 123 (51.2) |
| No | 29 (47.5) | 29 (49.2) | 29 (48.3) | 30 (50.0) | 117 (48.8) |
| Comorbidity | | | | | |
| Present | 11 (18.0) | 9 (15.3) | 8 (13.3) | 8 (13.3) | 36 (15.0) |
| Absent | 50 (82.0) | 50 (84.7) | 52 (86.7) | 52 (86.7) | 204 (85.0) |

BMI, body mass index; LPV/r, lopinavir-ritonavir; UCLH, University College London Hospital.

As detailed in Fig 1, 13 participants withdrew from the trial and a further 28 discontinued medication but provided samples for analysis. Predominantly this was due to toxicity that occurred disproportionately in arms including lopinavir-ritonavir (see Safety below). Overall, 224 participants (93.3%) were included in the ITT analysis and 208 participants (86.7%) in the mITT analysis of the primary outcome excluding those with undetectable viral loads at both Day 1 and Day 5.

## Effect of favipiravir, lopinavir-ritonavir, or combination therapy on SARS-CoV-2 viral load

The primary outcome was SARS-CoV-2 viral load at Day 5 of therapy accounting for baseline viral load. Fig 2 and Table 3 present summary data for the ITT and mITT cohorts, while S1 Table presents summary viral loads and S2 Fig displays results at participant level. In the primary analysis, there was no significant effect of any treatment arm on viral load: change in viral load versus placebo for favipiravir monotherapy −0.57 $\log_{10}$ copies/mL (95% confidence interval (CI) −1.21 to 0.07, $p$ = 0.08), for lopinavir-ritonavir monotherapy −0.18 $\log_{10}$ copies/mL (95% CI −0.82 to 0.46, $p$ = 0.58). There was no significant interaction between favipiravir

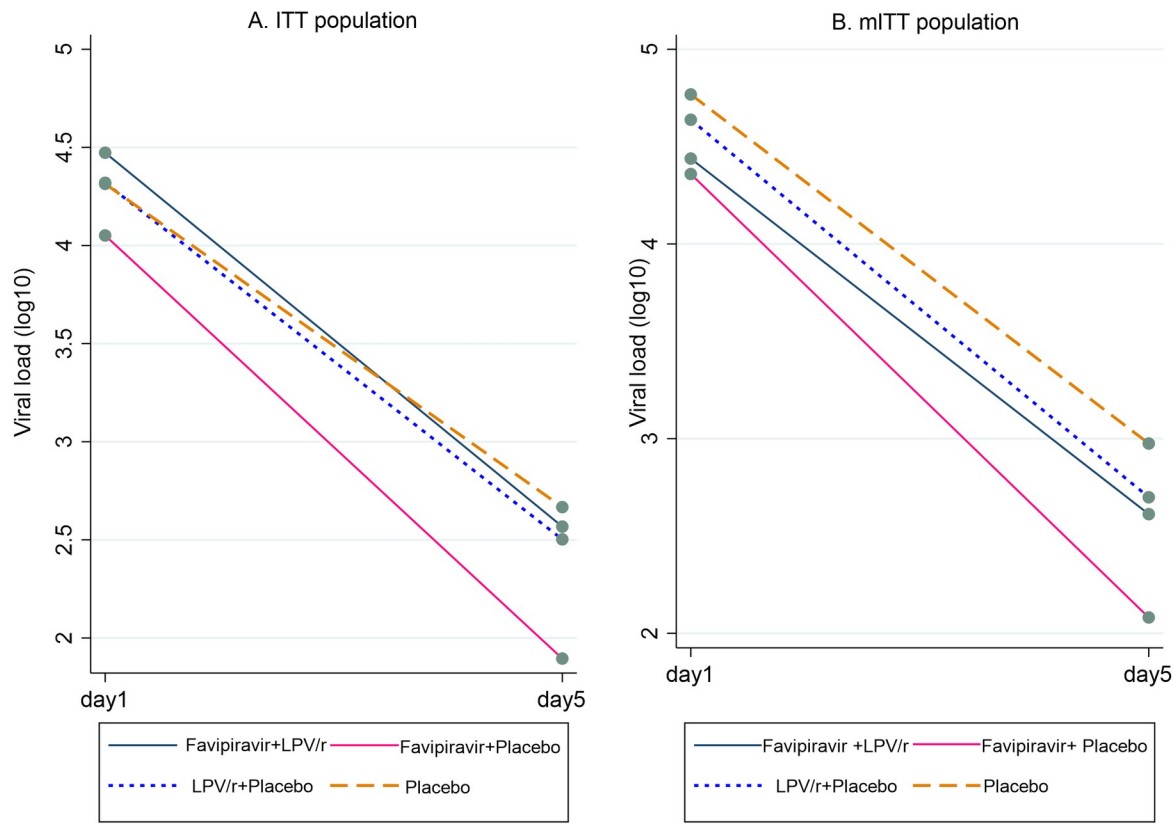

**Fig 2.** Mean $\log_{10}$ SARS-CoV-2 viral load at baseline (Day 1) and Day 5 per treatment arm in (A) the full ITT population and (B) the mITT population, excluding participants with negative viral load at baseline and Day 5. ITT, intention-to-treat; LPV/r, lopinavir-ritonavir; mITT, modified intention-to-treat; SARS-CoV-2, Severe Acute Respiratory Syndrome Coronavirus 2.

**Table 3. Primary outcome analysis: SARS-CoV-2 viral load at Day 5 accounting for baseline viral load.**

| | N | Favipiravir+Placebo (Main effect) | | LPV/r+Placebo (Main effect) | | Interaction Favipiravir +LPV/r | |
|---|---|---|---|---|---|---|---|
| | | Coefficient (95% CI) | *p*-value | Coefficient (95% CI) | *p*-value | Coefficient (95% CI) | *p*-value |
| **Primary outcome** | | | | | | | |
| ITT population | 224 | −0.57 (−1.21, 0.07) | 0.08 | −0.18 (−0.82, 0.46) | 0.58 | 0.59 (−0.32, 1.50) | 0.20 |
| Modified ITT population | 208 | −0.59 (−1.29, 0.11) | 0.10 | −0.18 (−0.87, 0.51) | 0.61 | 0.65 (−0.33, 1.63) | 0.19 |
| **Adjusted analyses of primary outcome** | | | | | | | |
| Adjusted for minimisation factors | 224 | −0.57 (−1.16, 0.02) | 0.06 | −0.14 (−0.73, 0.45) | 0.65 | 0.62 (−0.22, 1.46) | 0.15 |
| Adjusted for minimisation factors, symptom duration, antibody status | 222 | −0.65 (−1.23, −0.07) | 0.03 | −0.09 (−0.66, 0.49) | 0.76 | 0.66 (−0.16, 1.48) | 0.11 |
| **Mixed model analysis—At day 5** | | | | | | | |
| ITT population | 235 | −0.57 (−1.14, 0.01) | 0.05 | −0.24 (−0.81, 0.34) | 0.43 | 0.65 (−0.17, 1.47) | 0.12 |
| Adjusted for minimisation factors, symptom duration, antibody status | 233 | −0.63 (−1.17, −0.08) | 0.02 | −0.15 (−0.69, 0.40) | 0.60 | 0.65 (−0.11, 1.42) | 0.10 |

CI, confidence interval; ITT, intention-to-treat; LPV/r, lopinavir-ritonavir.

and lopinavir-ritonavir but the coefficient was numerically in the direction of antagonism (interaction coefficient: 0.59 $\log_{10}$ copies/mL, 95% CI −0.32 to 1.50, $p = 0.20$).

For favipiravir monotherapy, we observed similar effect sizes after adjustment for minimisation factors or for a potential effect of the delta variant of SARS-CoV-2 ($p = 0.06$). However, adjusting for the minimisation factors as well as symptom duration and antibody status, a stronger effect was noted (−0.65 $\log_{10}$ copies/mL [95% CI −1.23 to −0.07], $p = 0.03$). Following the same adjustment strategy and conditioning on baseline viral load, the mixed model analysis indicated a similar effect of favipiravir monotherapy (−0.63 $\log_{10}$ copies/mL [95% CI −1.17 to −0.08], $p = 0.02$; Table 3).

The proportion of participants with undetectable viral load at Day 5 was higher in the favipiravir monotherapy arm (odds ratio (OR) of being undetectable 2.47 [95% CI 1.08 to 5.65, $p = 0.03$]) but there was no effect of other treatment arms (Table 4).

In a post hoc supportive analysis (i.e., these were not predefined subgroups), we observed a significant interaction ($p = 0.03$) between treatment with favipiravir and baseline viral load levels (above or below the median level of 4.56 $\log_{10}$ copies/mL). In the low viral load group, there was no difference in Day 5 viral load between the treatment arms. However, in the high viral load group, favipiravir monotherapy was associated with a reduced viral load compared to placebo at Day 5 (difference 1.30 $\log_{10}$ copies/mL [95% CI 0.30 to 2.29]; Fig 3 and Table 5).

We also analysed results according to prespecified subgroups (vaccination status, antibody status, and duration of symptoms before commencing treatment ($\leq$5 days versus >5 days)) but did not observe any differences between treatments across these subgroups (Table 5).

Finally, we plotted average viral load in the ITT population (also dividing into high and low baseline viral load groups) and proportion with undetectable viral load per day of treatment (S3 and S4 Figs). Broadly, similar patterns were observed throughout the treatment course. Of note, we observed steeper decline of viral load in vaccinated or antibody-positive participants, with somewhat lower baseline viral loads in the latter, regardless of treatment arm (S5 and S6 Figs).

## Adverse events and safety reporting

A total of 518 adverse events were reported in 191 (80%) participants, of which 295 (57%) events were considered related to the treatment. The proportion of participants with treatment-related events was greater in those receiving lopinavir-ritonavir monotherapy (93%) and favipiravir plus lopinavir-ritonavir (88%) compared to those receiving favipiravir monotherapy (46%) and placebo (35%). The odds of experiencing a related event were significantly higher in the lopinavir-ritonavir arm compared to placebo (OR 16.0 [95% CI 4.27 to 60.0], $p < 0.0001$). Specifically, the occurrence of diarrhoea and nausea was higher in arms containing lopinavir-ritonavir. As detailed above, more participants in arms containing lopinavir-ritonavir discontinued treatment. Adverse events are summarised in S2 Table.

We also measured liver function tests at Day 1 and Day 7 (S3 Table and S7 Fig). Median levels for all parameters were within the normal range at both time points with minimal change during treatment. No clinically significant hepatitis or other hepatotoxicity was observed, but a

**Table 4. Odds ratios of achieving undetectable viral load (Ct $\geq$40) by Day 5.**

| | Sample size* | Placebo | Favipiravir+ Placebo (Main effect) | | | LPV/r+Placebo (Main effect) | | | Interaction Favipiravir+LPV/r | | |
|---|---|---|---|---|---|---|---|---|---|---|---|
| | | N (%) | N (%) | OR (95% CI) | *p*-value | N (%) | OR (95% CI) | *p*-value | N (%) | OR (95% CI) | *p*-value |
| Undetectable viral load | 203 | 14 (26.9) | 25 (46.3) | 2.47 (1.08, 5.65) | 0.03 | 17 (30.4) | 1.29 (0.55, 3.00) | 0.56 | 20 (35.7) | 0.52 (0.16, 1.66) | 0.27 |

* Patients included in this analysis had a detectable viral load at baseline and saliva sample available at Day 5.

CI, confidence interval; LPV/r, lopinavir-ritonavir; OR, odds ratio.

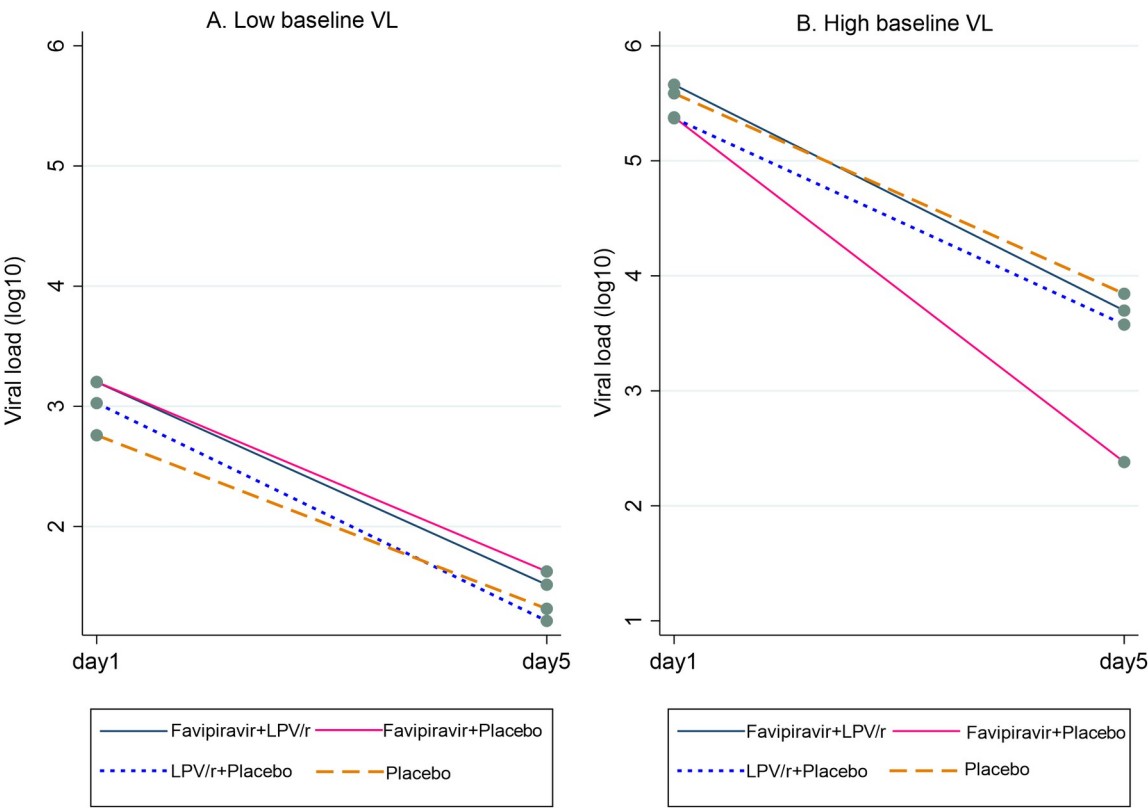

**Fig 3.** Mean $\log_{10}$ SARS-CoV-2 viral load at baseline (Day 1) and Day 5 per treatment arm in (A) participants with baseline viral load below or equal to the median level for the entire cohort and (B) participants with baseline viral load above the median level for the entire cohort. LPV/r, lopinavir-ritonavir; SARS-CoV-2, Severe Acute Respiratory Syndrome Coronavirus 2; VL, viral load.

minority of participants had a mild transaminitis before or during treatment. Participants with abnormal tests had repeat samples on Day 14 (S7 Fig).

As expected, serum uric acid levels significantly increased in the arms containing favipiravir (OR for elevated uric acid level in favipiravir monotherapy arm 18.8 [95% CI 4.2 to 84.8], $p < 0.0001$) after 7 days of treatment. However, the high levels were not sustained at Day 14.

There were 3 serious adverse events during the trial, all were hospitalisation due to progression of COVID-19. One event was seen in each of the lopinavir-ritonavir monotherapy, favipiravir monotherapy, and combination treatment arms. One participant (in the favipiravir monotherapy arm) was admitted to intensive care. There were no deaths in the study.

### Favipiravir drug levels at Day 7 are lower when administered together with lopinavir-ritonavir

All participants still taking trial medication and who were seen on Day 7 had blood samples taken pre-dose and 30 to 60 minutes post-dose for measurement of favipiravir drug levels. Assays were run on samples from 31 participants in the favipiravir monotherapy arm and 28 participants in the combination arm. As shown in Fig 4, favipiravir levels at both trough and peak were significantly lower in the combination treatment arm than in the favipiravir monotherapy arm. Of note, only a minority of participants achieved levels close to the EC90. S4 Table summarises demographic data on this cohort of participants, which did not differ between the arms or from the overall characteristics of the participants randomised to these arms.

**Table 5. Subgroup analyses for primary outcome according to vaccination status, duration of symptoms, baseline antibody status, and baseline viral load.**

| | N | Placebo | Favipiravir+Placebo (Main effect) | | | LPV/r+Placebo (Main effect) | | | Interaction Favipiravir+LPV/r | | |
|---|---|---|---|---|---|---|---|---|---|---|---|
| | | N | N | Coefficient (95% CI) | Interaction p-value | N | Coefficient (95% CI) | Interaction p-value | N | Coefficient (95% CI) | Interaction p-value |
| **Vaccinated** | | | | | | | | | | | |
| Yes | 117 | 29 | 28 | −0.71 (−1.66, 0.24) | 0.67 | 29 | 0.15 (−0.79, 1.09) | 0.32 | 31 | 0.90 (−0.43, 2.23) | 0.57 |
| No | 107 | 29 | 28 | −0.41 (−1.09, 0.27) | | 26 | −0.45 (−1.14, 0.24) | | 24 | 0.36 (−0.64, 1.35) | |
| **Days from symptom onset** | | | | | | | | | | | |
| ≤5 days | 148 | 35 | 38 | −0.37 (−1.17, 0.44) | 0.55 | 35 | 0.02 (−0.79, 0.84) | 0.50 | 40 | 0.48 (−0.65, 1.61) | 0.93 |
| >5 days | 75 | 22 | 18 | −0.80 (−1.86, 0.26) | | 20 | −0.43 (−1.46, 0.60) | | 15 | 0.42 (−1.13, 1.97) | |
| **Baseline antibody status** | | | | | | | | | | | |
| Negative | 80 | 23 | 20 | −0.06 (−0.75, 0.63) | 0.27 | 20 | −0.13 (−0.81, 0.55) | 0.98 | 17 | −0.09 (−1.10, 0.91) | 0.24 |
| Positive | 143 | 35 | 35 | −0.86 (−1.72, −0.01) | | 35 | −0.14 (−1.0, 0.72) | | 38 | 1.08 (−0.11, 2.28) | |
| **Baseline viral load** | | | | | | | | | | | |
| ≤ Median viral load | 117 | 27 | 36 | 0.12 (−0.72, 0.96) | 0.03 | 35 | −0.20 (−1.11, 0.70) | 0.94 | 29 | 0.09 (−1.13, 1.31) | 0.17 |
| >Median viral load | 107 | 31 | 20 | −1.30 (−2.29, −0.30) | | 30 | −0.13 (−1.01, 0.76) | | 26 | 1.28 (−0.09, 2.65) | |

CI, confidence interval; LPV/r, lopinavir-ritonavir.

## Secondary outcome measures

There was no difference in duration of fever between the arms, which was only observed in a minority of participants. There were also no differences between the arms in the proportion of participants with positive anti-spike antibody by Day 7, quantitative antibody levels, or the magnitude of change from Day 1. Key secondary outcomes are summarised in S5 Table.

## Discussion

The major finding of the FLARE trial is that, at the doses used, there is no clear evidence that either favipiravir monotherapy, lopinavir-ritonavir monotherapy, or favipiravir plus lopinavir-

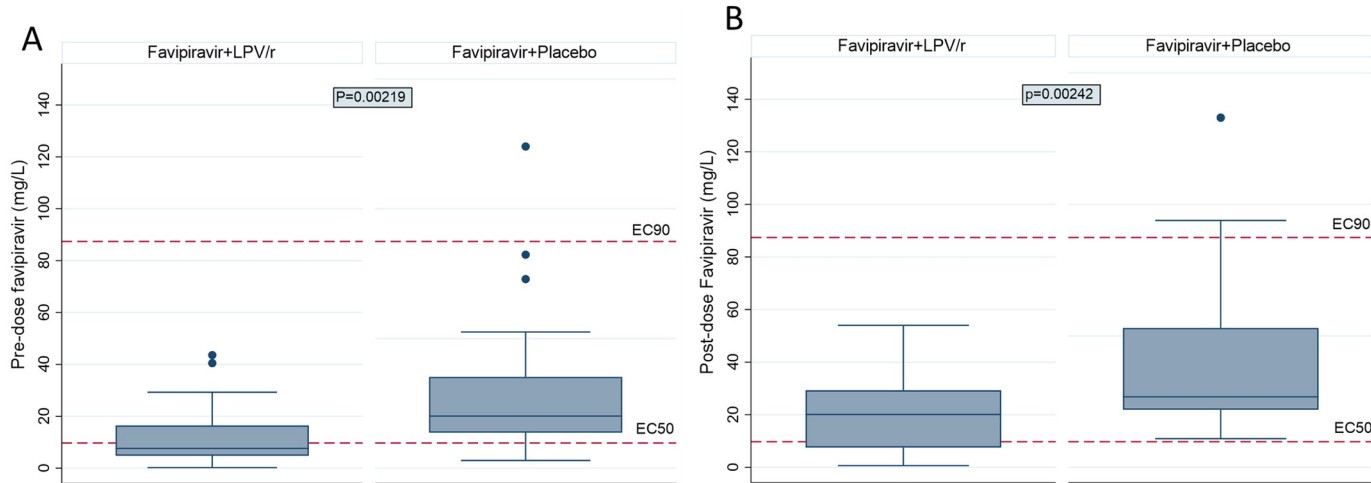

**Fig 4.** Plasma favipiravir concentration in the combination favipiravir+lopinavir-ritonavir (LPV/r) arm and the favipiravir+placebo arm on Day 7 (A) pre-dose (trough) and (B) 30–60 minutes post-dose (peak). Boxes represent IQR and whiskers represent 1.5*IQR. EC50: half maximal effective concentration. EC90, 90% maximal effective concentration. IQR, interquartile range.

ritonavir produce clinically worthwhile reductions in viral load in early treatment. FLARE provides insufficient evidence to take these therapies into Phase 3. However, further study of favipiravir may be warranted: In particular, dose escalation studies might identify more efficacious doses against SARS-CoV-2.

We found a numerically greater but nonsignificant reduction in viral load associated with favipiravir monotherapy in the primary analysis, while a post hoc fully adjusted mixed model, similar to that used to report the effect of other antivirals [5,22], was statistically significant (Table 3 and Fig 2). We also observed an increase in the proportion of patients with undetectable viral load compared to placebo, lopinavir-ritonavir, or combination therapy (Table 4 and S4 Fig). The effect was seen especially in those with higher baseline viral load, likely due to viral replication having slowed substantially in those with low viral load, limiting utility of antivirals [18,19]. However, this may point towards efficacy in a group with the most potential to benefit.

When the FLARE trial was designed in March 2020, we identified the imperative to generate high-quality Phase 2 proof of principle trial evidence on repurposed antivirals for early treatment of COVID-19, and this question remains important. The trial opened for recruitment in September 2020 but proceeded at only 2 sites as we did not receive research prioritisation in the UK via Urgent Public Health (UPH) status.

Based on in vitro data and early clinical reports, favipiravir was chosen as the most promising orally available agent. Due to uncertainty whether favipiravir monotherapy would be effective, the addition of lopinavir-ritonavir was proposed as an inexpensive, readily available protease inhibitor with evidence of some clinical effect against previous coronaviruses, and modest in vitro anti-SARS-CoV-2 activity.

Favipiravir is a ribosomal-dependent RNA polymerase (RdRp) inhibitor with a similar mode of action to molnupiravir. The magnitude of difference in viral load at Day 5 with favipiravir versus placebo in our trial (0.57 to 0.65 $\log_{10}$ copies/mL, depending on analysis used) was similar to that seen with molnupiravir (0.55 $\log_{10}$ copies/mL) at the highest dose tested in trials (800 mg twice daily) [22]: This agent has been reported to be clinically effective for early COVID-19. However, it remains to be seen whether molnupiravir monotherapy will retain clinical benefits in routine clinical practice. Favipiravir as monotherapy was well tolerated with relatively few adverse effects; in particular, we did not observe significant hepatotoxicity. A loading dose of 6,000 mg (2,400 mg given twice 8 hours apart followed by 1,200 mg) on Day 1, followed by 1,200 mg twice daily thereafter was well tolerated when used in Ebola [23] and thus higher doses may be feasible. High levels of uric acid were seen, which is a well-recognised side effect of favipiravir, but without obvious clinical consequence.

We chose the favipiravir dose used in influenza trials of 3,600 mg on Day 1 followed by 1,600 mg daily thereafter because simulations using pharmacokinetic data provided by Fujifilm Toyama Chemical Co. suggested we should expect to achieve 90% viral replication inhibition (along with a slight advantage in higher pre-dose trough levels if the maintenance dose was split 4 times per day rather than twice per day (S1 Fig)). However, upon measuring favipiravir pharmacokinetics on Day 7, we found levels around one third of our pretrial predictions and, perhaps more unexpectedly, significantly lower levels of favipiravir in the combination arm despite measurement being limited to those still taking IMP at this time point (Fig 4).

Our dosing simulations assumed linear pharmacokinetics and although there was a prior report of time-dependent reductions in levels seen in Ebola [23], we did not anticipate this with our dose regimen. However, data published after the start of FLARE indicate that favipiravir is likely to display time-dependent nonlinear pharmacokinetics at the doses used here [24], albeit intracellular concentrations with this dose regimen have been proposed to reach antiviral levels [25]. Nevertheless, time-dependent nonlinearity does not account for the lower

levels seen in the combination compared with monotherapy arm. While a cytochrome P450 mediated drug–drug interaction is not expected between favipiravir and lopinavir-ritonavir, possible explanations include lower favipiravir absorption associated with the gastrointestinal effects of lopinavir-ritonavir, or more unreported missed doses in the combination arm.

It remains possible that a concentration-dependent antiviral effect may nevertheless occur with the lower concentrations seen in FLARE, especially via mutagenesis. Viral sequencing work is ongoing to explore this possibility and a population pharmacokinetic-pharmacodynamic model is planned to investigate whether there is a concentration-response relationship with either viral load or mutagenesis. This model should identify the rationale for and doses to use in a future trial.

Favipiravir is in routine usage for COVID-19 in many countries, but existing trial data are mixed. Some small, open-label studies have indicated benefits in terms of clinical outcomes [26–29] or viral shedding [14,27]. However, other studies have indicated no clinically important benefit [30,31], including when given in early disease [32]. These studies were open label with heterogenous populations often including hospitalised patients, where antiviral treatment is expected to be less effective. Holubar and colleagues performed a double-blind randomised trial of favipiravir in asymptomatic or mildly symptomatic adults within 72 hours of a positive SARS-CoV-2 RT-PCR (median 5 days of symptoms) [15]. Among 116 patients, there was no difference in time to viral shedding cessation or symptom resolution. However, baseline Ct value (inversely related to viral load) tended to be lower while the change in Ct value between Days 1 to 7 tended to be greater in the favipiravir-treated arm.

By including a placebo-controlled lopinavir-ritonavir monotherapy arm, FLARE has demonstrated that this agent has no potential to reduce viral load, even with early treatment, and is poorly tolerated. As such, FLARE provides a strong rationale not to take lopinavir-ritonavir into Phase 3. We were able to reach this conclusion by exposing only 60 outpatients to lopinavir-ritonavir monotherapy. A similar design could have quickly ruled out other repurposed agents such as hydroxychloroquine.

An expected but problematic issue encountered with lopinavir-ritonavir was the frequency of side effects, especially gastrointestinal, leading to frequent discontinuation of treatment. We also encountered numerous potential drug–drug interactions, including with commonly prescribed medications such as budesonide and simvastatin, requiring exclusion of potential participants or modification/suspension of concomitant medications. These are important issues to consider with other ritonavir-boosted protease inhibitors (e.g., nirmatrelvir). Fortunately, data from trials so far have not indicated a high rate of discontinuation with nirmatrelvir-ritonavir [5]. Possible explanations include fewer side effects with nirmatrelvir compared to lopinavir, improved adherence among a high-risk group with more potential for clinical benefit and better adherence to a twice daily regime. However, this will need to be monitored in "real world" settings.

As the recruitment period for FLARE coincided with the successful UK vaccine roll-out, the Trial Steering Committee decided to include participants who had received a vaccine. Regardless of treatment arm, rate of viral load decay tended to be higher in participants who were vaccinated or antibody-positive at baseline.

Our study has some limitations. The recruited cohort was relatively young and healthy with lower rates of comorbidities and obesity than often seen in the hospitalised population with COVID-19. Approximately 18% of participants were of non-white ethnicity that is similar to the UK population but may be less reflective of those with poor outcome. However, the trial was designed predominantly to look at viral load rather than clinical outcomes and therefore these factors were less important. Our participants had lower baseline viral loads than many reported elsewhere in the literature. This may have related in part to the use of saliva, but saliva

kits were convenient for participants and allowed us to standardise the volume collected. In other reports, viral load has been reported to be either higher [33] or lower [34] in saliva compared to nasopharyngeal swab but generally concordance is very good [33–35]. We were unable to perform viral culture or infectivity assays that may have provided useful additional information. For logistical reasons, we were unable to obtain samples for pharmacokinetics on every participant in the study. Due to the lack of UK UPH status, recruitment took longer than anticipated, and the results of FLARE are available only after the identification of other oral antiviral agents for COVID-19. However, they retain importance for low- and middle-income countries where favipiravir is already in routine use and where molnupiravir or nirmatrelvir-ritonavir may be prohibitively expensive.

In conclusion, our results do not support routine usage or Phase 3 trials of favipiravir or lopinavir-ritonavir at the doses investigated. There may be some effect of favipiravir when used for early treatment of COVID-19, especially in those with high baseline viral load, but further investigation is needed regarding dosage or additive antiviral medication. Another relatively small study would be sufficient to establish this. We have conclusively demonstrated the ineffectiveness of lopinavir-ritonavir even in early disease and have identified a new drug interaction between favipiravir and lopinavir-ritonavir with the latter apparently lowering plasma levels of the former. These results have important implications for the global efforts against COVID-19.

## Supporting information

**S1 Appendix. List of FLARE Investigators.**
(DOCX)

**S2 Appendix. Sample size simulation code in R.**
(DOCX)

**S1 Table. Summary viral load data per arm.**
(DOCX)

**S2 Table. Summary of adverse events.**
(DOCX)

**S3 Table. Serum liver function tests and uric acid at Day 1 and Day 7.**
(DOCX)

**S4 Table. Baseline characteristics of the cohort with pharmacokinetic measurements.**
(DOCX)

**S5 Table. Summary of key secondary outcomes.**
(DOCX)

**S1 Fig. Pharmacometric modelling of predicted plasma concentrations for favipiravir and lopinavir-ritonavir at the doses used in the FLARE trial, presented in relation to the half maximal effective concentration (EC50) and 90% maximal effective concentration (EC90).** Simulations are presented for a twice daily (BD) dosing regime and four times daily (QDS) dosing regime.
(TIF)

**S2 Fig.** Log10 SARS-CoV-2 viral load at baseline (Day 1) and Day 5 presented per participant for (A) favipiravir+lopinavir-ritonavir (LPV/r), (B) favipiravir+placebo, (C) LPV/r + placebo and (D) placebo only.
(TIF)

**S3 Fig.** Mean log10 SARS-CoV-2 viral load per treatment arm on each day of treatment in (A) the entire cohort, (B) participants with baseline viral load below or equal to the median level and (C) participants with baseline viral load above the median level.
(TIF)

**S4 Fig. Proportion of participants with detectable viral load at baseline who had undetectable viral load (Ct $\geq$40) on each subsequent day of treatment, per treatment arm.** Underlying data are presented in the accompanying table.
(TIF)

**S5 Fig. Mean log10 SARS-CoV-2 viral load per treatment arm on each day of treatment and per study arm presented according to vaccination status.** LPV/r, lopinavir-ritonavir.
(TIF)

**S6 Fig. Mean log10 SARS-CoV-2 viral load per treatment arm on each day of treatment and per study arm presented according to baseline antibody status.** LPV/r, lopinavir-ritonavir.
(TIF)

**S7 Fig.** (A) Serum ALT concentration, (B) serum AST concentration and (C) serum uric acid concentration at Day 1, Day 7 and Day 14 according to treatment arm. Blood tests were usually only taken at Day 14 if abnormal at Day 7. Boxes represent IQR and whiskers represent 1.5*IQR. ALT, alanine aminotransferase; AST, aspartate aminotransferase; IQR, interquartile range.
(TIF)

**S1 Consort Checklist. CONSORT Checklist.**
(DOC)

**S1 Statistical Analysis. Statistical Analysis Plan, v1.0, 4 November 2021.**
(DOCX)

**S1 Trial Protocol. Protocol v6.0, 8 June 2021.**
(DOCX)

## Acknowledgments

The authors would like to acknowledge: the Agile Lighthouse team within UKHSA for assistance with recruitment and Fujifilm Toyama Chemical Co. who provided favipiravir and favipiravir placebo free of charge. We also acknowledge Professor Chris Frost as a non-independent member of the Trial Steering Committee and Dr. Mak Wenyao for intellectual contribution.

## Author Contributions

**Conceptualization:** David M. Lowe, Hakim-Moulay Dehbi, Nick Freemantle, Judith Breuer, Joseph F. Standing.

**Formal analysis:** Kashfia Chowdhury, Akosua A. Agyeman, Hakim-Moulay Dehbi, Nick Freemantle, Joseph F. Standing.

**Funding acquisition:** David M. Lowe, Nick Freemantle, Judith Breuer, Joseph F. Standing.

**Investigation:** David M. Lowe, Li-An K. Brown, Stephanie Davey, Philip Yee, Divya Shah, Alexander Lennon, Abhulya Rai, Anna Checkley, Nicola Longley.

**Methodology:** David M. Lowe, Hakim-Moulay Dehbi, Nick Freemantle, Judith Breuer, Joseph F. Standing.

**Project administration:** Li-An K. Brown, Felicia Ikeji, Amalia Ndoutoumou.

**Writing – original draft:** David M. Lowe, Joseph F. Standing.

**Writing – review & editing:** David M. Lowe, Li-An K. Brown, Kashfia Chowdhury, Stephanie Davey, Philip Yee, Felicia Ikeji, Amalia Ndoutoumou, Divya Shah, Alexander Lennon, Abhulya Rai, Akosua A. Agyeman, Anna Checkley, Nicola Longley, Hakim-Moulay Dehbi, Nick Freemantle, Judith Breuer, Joseph F. Standing.

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
