## [Editor Report · Decision Letter 0]

13 Apr 2022

Dear Dr Lowe, 

Thank you for submitting your manuscript entitled "Favipiravir, lopinavir-ritonavir or combination therapy (FLARE): a randomised, double blind, 2x2 factorial placebo-controlled trial of early antiviral therapy in COVID-19" for consideration by PLOS Medicine.

Your manuscript has now been evaluated by the PLOS Medicine editorial staff and I am writing to let you know that we would like to send your submission out for external peer review.

Please re-submit your manuscript within two working days, i.e. by Apr 15 2022 11:59PM.

Kind regards,

Louise Gaynor-Brook, MBBS PhD

Senior Editor

PLOS Medicine

---

## [Decision Letter · Decision Letter 1]

27 Jul 2022

Dear Dr. Lowe,

Thank you very much for submitting your manuscript "Favipiravir, lopinavir-ritonavir or combination therapy (FLARE): a randomised, double blind, 2x2 factorial placebo-controlled trial of early antiviral therapy in COVID-19" (PMEDICINE-D-22-01160R1) for consideration at PLOS Medicine. 

[LINK]

In light of these reviews, I am afraid that we will not be able to accept the manuscript for publication in the journal in its current form, but we would like to consider a revised version that addresses the reviewers' and editors' comments. Obviously we cannot make any decision about publication until we have seen the revised manuscript and your response, and we plan to seek re-review by one or more of the reviewers. 

We expect to receive your revised manuscript by Aug 17 2022 11:59PM. Please email us (plosmedicine@plos.org) if you have any questions or concerns.

We look forward to receiving your revised manuscript. 

Best wishes,

Philippa

Dr. Philippa Dodd, MBBS MRCP PhD

PLOS Medicine

plosmedicine.org

GENERAL

Please complete the CONSORT checklist and ensure that all components of CONSORT are present in the manuscript, including [how randomization was performed, allocation concealment, blinding of intervention, definition of lost to follow-up, power statement]. When completing the checklist, please use section and paragraph numbers, rather than page numbers.

A pre-specified statistical analysis plan is mentioned in the methods section (line 183). Please include the relevant prospectively written document with your revised manuscript as a Supporting Information file to be published alongside your study, if accepted and cite it in the Methods section. A legend for this file should be included at the end of your manuscript. Please also include the study protocol document.

Please remove COI/funding source/ info from the end of the manuscript and instead include in the manuscript submission form 

ABSTRACT

Please structure your abstract using the PLOS Medicine headings (Background, Methods and Findings, Conclusions). Please combine the Methods and Findings sections into one section, “Methods and findings”. 

In the last sentence of the Abstract Methods and Findings section, please describe the main limitation(s) of the study's methodology.

METHODS and FINDINGS

When a p value is given, please specify the statistical test used to determine it.

FIGURES & TABLES

Please consider avoiding the use of red and green in order to make your figures more accessible to those with colour blindness

Please define the abbreviations in Figure 3 (VL), figure 4 (EC50, EC90), suppl figure 5 (LPV/r), supp figure 6 (the treatment arms are labelled differently than other figures/tables throughout the manuscript i.e. favipira Vs favipiravir. Please ensure congruency and include abbreviations as necessary)

Please define the abbreviation in table 2 (BMI), table 3 (ITT, CI), table 4 (CI), table 5 (CI), supp table 2 (IQR), supp table 3 (Pk)

Please indicate in the figure caption the meaning of whiskers in Figure 4

Please indicate in the figure caption the meaning of whiskers in supp figure 6

DISCUSSION

Please present and organize the Discussion as follows: a short, clear summary of the article's findings; what the study adds to existing research and where and why the results may differ from previous research; strengths and limitations of the study; implications and next steps for research, clinical practice, and/or public policy; one-paragraph conclusion.

Comments from the reviewers:

Reviewer #1: Statistical Review

Lowe and colleagues present the results of a 2x2 factorial designed study of early antiviral treatment in COVID-19. This review considers the use of statistics in the paper.

Overall, these are very good, but I do have a few minor comments.

The modelling approach is reasonable, and the authors apply a number of sensitivity analyses, but I would like to have seen the results of models fitted without interaction terms; this could be done in a supplementary table.

In the text, the primary analysis results are reported as reductions in the treated arms relative to placebo/placebo, though the confidence intervals quoted are for the differences (i.e. the opposite sign to the point estimates), which is slightly confusing.

The main conclusion that there is no clear evidence that either treatment produces clinically relevant reductions in viral load, seems a little negative. A clinically important reduction is quoted as being in the range 0.5 to 1.0 log10 copies per ml, and the main effect of favipiravir is 0.57 in the primary analysis, or more in the supporting analyses, and much more in the subgroup with higher baseline viral load. Saying that, the authors do discuss these points.

There is mention of a pre-specified statistical analysis plan, but it is not referenced, and not included in the supplementary materials.

Reviewer #2: Lowe et al report the results of a blinded randomised clinical trial for COVID patients of favipiravir, LPV/r, the combination, and placebo. The primary outcome was salivary viral load at day 5 (adjusted for baseline level). This was a phase 2a trial. There was no statistically significant change in viral load compared to placebo for any of the treatment arms in the primary analysis.

Overall, this is a well written manuscript that reports on the trial as planned in the trial protocol and as registered on clinicaltrials.gov. The use of placebo is a particular strength. It provides an example of a well conducted phase 2 trial that can inform the design of future trials. As the authors state in the discussion more trials such as this may have helped to guide which agents should be included in larger phase 3 trials during a pandemic. Unfortunately, the time taken to complete this study also provides an example that such earlier phase trials need to be conducted and reported quickly to make such an impact. The reality is that the results from FLARE are unlikely to have great impact on the therapeutic landscape or subsequent trials given the alternative options in mid 2022 - at least in high income countries. One suggestion is that in the discussion the authors could consider the issue of whether favipiravir should still be used or trialled in lower middle income countries that may not have access to monoclonals or paxlovid and molnupiravir.

I would like to see details on whether all randomised participants had confirmed SARS-CoV-2 - the first inclusion criteria appears to only require consistent symptoms. The protocol suggests that they all were swabbed for a diagnosis, but I think they could be randomised prior to the results. Can probably infer this from the figures, but should report at start of results.

In terms of the reporting of the primary outcome, I would have liked to have seen more detail in the main manuscript. Specifically in table 3, reporting the coefficient was not intuitively understood. To me it would be clearer to see the baseline and day 5 viral loads (I know this is in figure 2, but the summary mean numbers could be reported in the text or table) in each of the 4 groups, the change in viral load in the 4 groups, and then the coefficient which represents the difference in change between the 3 active groups and the placebo group (if I have understood the analysis). Figure 2 and 3 and suppl figure 2 are certainly helpful as well. Table 4 is also helpful in seeing the raw numbers in terms of numerator and denominator. At line 234-235, I'm not sure it is appropriate to say the pre-defined threshold for significance. I haven't seen the Statistical analysis plan, but I would have thought the pre-defined threshold related to the primary analysis, and not to the various additional analyses that were performed. I think better to just state the numbers. Same point when this is discussed in the discussion at line 313-314. At line 240-245, was this 'post-hoc supportive analysis' in pre-defined sub-groups? Would be good to be clear about that. E.g, 'In the post-hoc supportive analysis (i.e., this was not a pre-defined sub-group analysis)…'

For clarity and ease of use of data for meta-analyses etc, I suggest all the secondary outcomes be reported in a table (excluding the PK-PD modelling aspects). I can get this data from the manuscript text, but I would encourage clear reporting of all the secondary outcomes.

Line 324 - the clinical trial results for molnupiravir have now been reported.

Line 354-358 - rather than using 'this' in some of the sentences, it would be clearer to directly refer to 'LPV-ritonavir'.

Line 369-378 - the paragraph could come earlier in the discussion. I suggest grouping all the favipiravir paragraphs together.

Limitations wise - a discussion of timeliness of results and therefore ability to impact on subsequent trials or practice could be considered in this section.

Reviewer #3: The authors report on the FLARE study which was a Phase 2, proof of principle, randomised, placebo-controlled, 2x2 factorial, double-blind trial of outpatients with early COVID-19 (within 7 days of symptom onset) at two sites in the United Kingdom. The study consisted of 4 arms; 

1) favipiravir + lopinavir-ritonavir.

2) favipiravir plus lopinavir-ritonavir placebo.

3) lopinavir-ritonavir plus favipiravir placebo.

4) both placebos. 

The primary outcome was viral load relative to baseline on day 5.

Abstract

The favipiravir alone arm demonstrated the greatest reductions in viral load but the confidence intervals were large ( 1.21 to 0.07 log10 reductions, p=0.08) leading to a non significant result. 

Methods

The patient population are patients diagnosed with coronavirus within 7 days of symptom onset who were managed in the outpatient setting. The authors state ambulatory patients in the methods and then in exclusion mention participants were being treated as a hospital inpatient for any condition. They should just make this clear in the abstract and methods that the trial participants were treated in the outpatient/community setting (hence their population is classed as a non severe patient group).

The viral Laos was measured in daily saliva samples. Is there sufficient data to show how it correlates in terms of viral load with nasopharyngeal samples. 

Statistical analyses

The study is inadequately powered but at the time of setting up the study in the early phase of the pandemic there was insufficient information and the authors have made reasonable attempts at estimating effect size. 

Results

Dominant population that has been recruited was under the age of 55 years with average age of 40 years. Hence, in this healthy population the placebo groups may have also more effectively cleared the virus and hence reducing the difference between the treatment and placebo group. Furthermore, almost 50% of patients recruited had received 2 doses of the vaccine. This may have further reduced the effective difference in viral clearance between the treat,ent and placebo groups. Have the authors factored age and vaccination into their linear mixed model. It is very likely that a lot of the adjustments are post hoc and the authors need to be explicit in which factors were planned a priori and which factors were post hoc eg effect of delta variant, effect of vaccination if considered.

Do the authors have an explanation why only 50% of patients had drug levels measured?

Viral pcr: did favipiravir have any effect on the viral sequence on day 5 compared to the other therapies. As favipiravir is a ribosomal-dependent RNA polymerase (RdRp) inhibitor we would anticipate multiple mutations in the virus. The authors mention they are performing viral sequencing and that data would significantly strengthen this study.

Reviewer #4: Overall summary: This phase 2 proof of principle, placebo controlled RCTwith 2x2 factorial double blind trial of OP with early COVID-19 from 2 sites in England using favipiravir, favipiravir and LPV-ritonavir, favipiravir plus LPV-ritonavir placebo, and both placebo using SARS-CoV2 VL at day 5 obtained through saliva as an outcome was safe, but with frequent side effects, most notably GI. There was no clinically significantly reduced VL in the primary analysis of the study drug. 

Main strengths: The analysis was done during the pandemic when there were substantial challenges in patient recruitment, research staffing and changes in standards of care. It was also very difficult to bring outpatients in for research studies in a safe way due to limitations on transportation and health care facility rules. The authors should be commended for conducting an outpatient trial during this time period due to these substantial barriers. 

Potential points to further explore: 

It is unclear why patients who were asymptomatic were included in the trial, as they would be less likely to show an effect (ie they likely had lower viral loads to begin with) do the results change by excluding those? 

In the demographics table, the authors indicated the number of doses received of the vaccine. It might be better to classify whether the participants who were vaccinated, were fully vaccinated or partially vaccinated (ie were they more than 14 days out from their most recent/final vaccination)? How did some patient receive 3 vaccines?

In the minimization table participants are labelled as vaccinated or not. Would redo this analysis with fully/partially/not vaccinated. 

The authors note that there was no significant reduction in VL in the primary analysis. The difficulty with the analysis is that the samples obtained were in the saliva, which is more convenient, but less likely to show an effect than a nasal swab. This is noted in the limitations, but significantly impacts the results of the study. 

A notable issue is the significant number of withdrawals/discontinuations of the groups taking LPV-r due to GI side effects, and the substantial drug-drug interactions. Appropriately the authors discuss the issues with using ritonavir for boosting; however this is being done fairly successfully with nirmatrelvir in 2022. Do the authors have any speculation for the differences between drop out rates for paxlovid real world data and for the LPV-r arms in this trial?

As noted in the limitations, this study population was young and very healthy. Would specifically note that there were few minorities, low rates of comorbidities and obesity, and over half were vaccinated, which could have affected the study effect. And make it less representative of those most affected by the COVID-19 pandemic. As such, would note this in the limitations. The authors do not report any data on resolution of symptoms other than duration of fever. It would be helpful to identify what symptoms patients had at the start of the trial and at the end to identify how sick they were. Presumably none of the patients were hospitalized, but that should be explicitly noted.

[LINK]

---

## [Decision Letter · Decision Letter 2]

26 Sep 2022

Dear Dr. Lowe,

Thank you very much for re-submitting your manuscript "Favipiravir, lopinavir-ritonavir or combination therapy (FLARE): a randomised, double blind, 2x2 factorial placebo-controlled trial of early antiviral therapy in COVID-19" (PMEDICINE-D-22-01160R2) for review by PLOS Medicine.

I have discussed the paper with my colleagues and the academic editor and it was also seen again by xxx reviewers. I am pleased to say that provided the remaining editorial and production issues are dealt with we are planning to accept the paper for publication in the journal.

[LINK]

We look forward to receiving the revised manuscript by Oct 03 2022 11:59PM.   

Sincerely,

Philippa Dodd, MBBS MRCP PhD

PLOS Medicine

pdodd@plos.org

plosmedicine.org

Requests from Editors:

GENERAL

The Data Availability Statement (DAS) requires revision. For each data source used in your study: 

In the first 1/3-1/2 of the manuscript you use square brackets to sign post tables and/or figures etc, in the latter part circular brackets. Suggest the use of circular brackets throughout reserving square brackets for in-text reference call-outs and for however duplication of brackets are necessary)

AUTHOR SUMMARY

Thank you for including an author summary I have made some suggestions for modifications as below, as we understand things (please note the requirement for bullet points):

Why was this study done? 

 * The FLARE trial aimed to discover whether existing oral antiviral drugs could reduce the viral load of the SARS-CoV-2 virus if given soon after symptoms started.

 * If effective this strategy could reduce the risk of hospitalisation and death from COVID-19. 

What did the researchers do and find? 

 * The researchers performed a clinical trial of two medications - favipiravir and lopinavir/ritonavir, testing them on their own and in combination. 

 * Combination therapies were less effective than favipiravir monotherapy, but many people taking lopinavir/ritonavir had gastrointestinal side effects and favipiravir drug levels were lower in the combination arm, possibly due to poor absorption.

 * SARS-CoV-2 viral loads were not significantly lower with any of the drug treatments after 5 days compared to placebo, although more people taking favipiravir had undetectable levels of the virus. 

What do these findings mean? 

 * None of these therapies should be used routinely at the current doses investigated.

 * Further studies investigating the effect of Favipiravir when administered at higher doses should be undertaken.

METHODS and RESULTS

Please indicate explicitly in the main manuscript text that consent was written.

TABLES

Table 3: please also provide unadjusted analyses

FIGURES

Suppl figure 5: the graphs are very small thus rather inaccessible to the reader, please revise

Comments from Reviewers:

Reviewer #1: Alex McConnachie, Statistical Review

I thank the authors for their responses to my original comments. I am generally happy with these.

Seeing the SAP is good, but the paper does not signpost it when it is referred to in the methods section. I had hoped that the SAP would provide more information about the sample size calculation/simulation, but the text in the SAP is very much the same as the paper. Sufficient detail needs to be given to allow someone to replicate what was done (e.g. assumed treatment effects, and SD of the primary outcome).

My guess is that the sample size calculation for the main effects relates to a model without interaction, essentially treating the study as two separate trials, where the analysis in relation to one randomisation treats the other randomisation as a stratification factor; this is why I prefer that the non-interaction analysis is reported somewhere in the paper. Saying that, it is true that in this case, the interaction model is more informative, given the magnitude of the interaction term. Plus, it is possible to work out the results without the interaction, given the information presented, so seeing these results is more for convenience, than a requirement.

Reviewer #2: I am satisfied that the authors have addressed the comments made on previous review. Well done on a valuable trial.

[LINK]

---

## [Editor Report · Decision Letter 3]

6 Oct 2022

Dear Dr Lowe, 

On behalf of my colleagues and the Academic Editor, Dr Amitabh Suthar, I am pleased to inform you that we have agreed to publish your manuscript "Favipiravir, lopinavir-ritonavir or combination therapy (FLARE): a randomised, double blind, 2x2 factorial placebo-controlled trial of early antiviral therapy in COVID-19" (PMEDICINE-D-22-01160R3) in PLOS Medicine.

We thank you for your considerate responses to previous editor and reviewer comments. There is just one further very minor revision that needs your attention prior to publication. From line 6 of the section entitled: “Effect of favipiravir, lopinavir-ritonavir or combination therapy on SARS-CoV-2 viral load” p-values are reported as “, p=0.08” and later as “p<0.0001”. Please check throughout and remove “=” where p-values are reported.

PRESS

Sincerely, 

Pippa

Philippa Dodd, MBBS MRCP PhD

pdodd@plos.org

PLOS Medicine